# Selective Suppression of Cell Growth and Programmed Cell Death-Ligand 1 Expression in HT1080 Fibrosarcoma Cells by Low Molecular Weight Fucoidan Extract

**DOI:** 10.3390/md17070421

**Published:** 2019-07-19

**Authors:** Kiichiro Teruya, Yoshihiro Kusumoto, Hiroshi Eto, Noboru Nakamichi, Sanetaka Shirahata

**Affiliations:** 1Faculty of Agriculture, Kyushu University, 744 Motooka, Nishi-ku, Fukuoka 819-0395, Japan; 2Graduate School of Bioresource and Bioenvironmental Sciences, Kyushu University, 744 Motooka, Nishi-ku, Fukuoka 819-0395, Japan; 3Daiichi Sangyo Co., Ltd., 6-7-2 Nishitenma, Kita-ku, Osaka 530-0047, Japan

**Keywords:** fucoidan, low molecular weight fucoidan extract, N-Ras, neuroblastoma-rat sarcoma, Cancer, programmed cell death-ligand 1, programmed cell death-ligand 2, human sarcoma cell line (HT1080 cells), human normal diploid fibroblast (TIG-1 cells)

## Abstract

Low molecular weight fucoidan extract (LMF), prepared by an abalone glycosidase digestion of a crude fucoidan extracted from *Cladosiphon novae-caledoniae* Kylin, exhibits various biological activities, including anticancer effect. Various cancers express programmed cell death-ligand 1 (PD-L1), which is known to play a significant role in evasion of the host immune surveillance system. PD-L1 is also expressed in many types of normal cells for self-protection. Previous research has revealed that selective inhibition of PD-L1 expressed in cancer cells is critical for successful cancer eradication. In the present study, we analyzed whether LMF could regulate PD-L1 expression in HT1080 fibrosarcoma cells. Our results demonstrated that LMF suppressed PD-L1/PD-L2 expression and the growth of HT1080 cancer cells and had no effect on the growth of normal TIG-1 cells. Thus, LMF differentially regulates PD-L1 expression in normal and cancer cells and could serve as an alternative complementary agent for treatment of cancers with high PD-L1 expression.

## 1. Introduction

Various types of cancers evade T-cell responses and host immunity via adaptive negative regulators or co-inhibitory (checkpoint) receptors, such as programmed cell death-1 (PD-1) and their respective ligands, programmed cell death-ligand 1 (PD-L1 or B7-H1), programmed cell death-ligand 2 (PD-L2), cytotoxic T-lymphocyte-associated antigen 4 (CTLA-4), and CD80 (B7.1) or CD86 (B7.2) [1,2]. These regulators react with each other to form an elaborative defense system or host immunity to kill pathogens and cancer cells. In order to acquire cytocidal function, a variety of immune cells, including resting T cells expressing PD-1, must be activated by two stimulating signals from an antigen-presenting cell (APC), as shown in Figure 1 [3,4,5]. In addition, resting T-cells express T-cell receptor (TCR) and CD28 on their cell surface which interact with the antigen-specific signal presented on the major histocompatibility complex (MHC) and CD80 (B7.1) or CD86 (B7.2) ligand expressed on the surface of an APC, respectively. Interactions of these receptor-ligand pairs are referred to as the early stage of T-cell activation [1,4]. The activated T-cells express CTLA-4, and as the binding affinity of CTLA-4 for CD80/86 is 500- to 2500-fold greater than that for CD28, CTLA-4 (negative regulator) competes with co-stimulatory CD28 signals to bind to CD80/86 [6,7].

Conversely, the interaction of PD-1 expressed on the T-cell surface with its ligand PD-L1 or PD-L2 on APC causes inhibition of T-cell activation, leading to reduced cell proliferation, and induces T-cell cytolysis [6,8]. While a balanced PD-1/PD-L1 interaction is a prerequisite for maintaining the normal T-cell response and assuring normal cell survival, overexpressed PD-L1 on cancer cells in the tumor microenvironment react with the PD-1 expressed on T cells and induce T-cell apoptosis, thereby facilitating tumor survival, progression, and evasion of the host tumor immune surveillance system [4,7,9]. Moreover, higher PD-L1 expression in many tumors corresponds to enhanced immune evasion thereby promoting tumor growth by suppressing the T-cell response [7,10,11,12]. Therefore, considering the PD-1/PD-L1 axis, the selective inhibition of PD-L1 expressed on cancer cells is expected to maintain the host defense system and consequently contribute towards cancer eradication.

PD-L1 expression is upregulated not only by some inflammatory cytokines (e.g., interferon-γ) but also by oncogenic activation of signaling pathways [6,12]. In one of the two main oncogenic signaling pathways, the extracellular epidermal growth factor (EGF) and TGF-α stimulate the EGF receptor (EGFR) which in turn activates wild type rat sarcoma oncogene homolog (Ras) to activate the raf pathway [14]. Moreover, in the second pathway, the constitutively active mutant Ras activates its immediately downstream factors, including phosphatidylinositol 3-kinase (PI3K), Rac1, and Raf which trigger a sequential activation of multiple downstream factors and eventually induce expressions of many genes such as RhoA/B, EGFR, VEGF, and PD-L1 in cancer cells [2,6,15,16,17].

Other lines of evidence have shown constitutive activation of mutant N-*ras* allele encoded in a human soft tissue sarcoma cell line, HT1080, which regulates Raf and RhoA pathways. Moreover, Raf-activated MEK, an intermediary transducer, is essential in the Ras signaling pathway but not in the PI3K/Akt pathway that includes nuclear factor-kappa B (NF-κB), resulting in the aggressive tumorigenic phenotype in HT1080 cells [18,19]. These data suggest that the Raf regulated pathway is closely associated with carcinogenesis, implying that this pathway regulates PD-L1 expression as one of the terminal steps in HT1080 cells. However, these studies do not address the involvement of Ras-regulated pathways in PD-L1 expression in HT1080 cells. As the PD-1/PD-L1 axis plays a major role in human cancers for immune evasion [17], it might be extremely valuable to cancer patients if prospective agents such as fucoidan devoid of side effects are available to regulate PD-L1 expression exclusively in cancer cells.

Fucoidan can be found mainly in various species of brown algae (brown seaweed) such as wakame (*Undaria pinnatifida*), kombu (*Saccharina japonica*), mozuku (*Cladosiphon okamuranus*), hijiki (*Sargassum fusiforme*), bladderwrack (*Fucus vesiculosus*), and hibamata (*Fucus evanescens*) [20,21]. Fucoidan is fucose containing sulfated polysaccharides exhibiting complicated chemical structures due to various combinations of fucose, uronic acids, galactose, and xylose. [22,23,24]. The structure of fucoidan from *Cladosiphon okamuranus* belonging to Genus *Cladosiphon* had been determined partly using the nuclear magnetic resonance (NMR) method [25] and we reported that this structure possess similar structural features as the fucoidan from *Cladosiphon novae-caledoniae* Kylin. It has been noted that Quantitative ^1^H-NMR (qNMR) analysis for the qualitative and quantitative characterization of metabolites may be applied for crude biological extracts such as fucoidan [26]. Just recently, the chemical structure of fucoidan from *Cladosiphon okamuranus* was determined using NMR analyses [27], as shown in Figure 2, and we now also refer to this structure as having a similar structural feature as fucoidan from *Cladosiphon novae-caledoniae* Kylin.

Fucoidan extracts usually show various sizes, such as 5100 kDa [28] and 1600 kDa [29], depending on the extraction methods and the kinds of seaweeds. Various molecular weight derivatives designated as low-molecular-weight fucoidan (LMWF: up to 40 kDa), intermediate-molecular-weight fucoidan (110–138 kDa), and high-molecular-weight fucoidan (300–330 kDa) [30], and other size classifications up to 10,000 kDa [31], have been prepared for biological and biochemical characterizations. In still other cases, many derivatives having molecular weights ranging between 7.6 kDa and 712 kDa have been assessed for their membrane permeability in vitro, and bioavailability in humans [32,33,34,35,36]. Apart from the above higher molecular weight derivatives, there are a group of fucoidan derivatives possessing much lower molecular weights designated as low molecular weight fucoidan (LMF) also called as oligo-fucoidan having MW of 500–800 Da [37], LMF from L. japonica (LMF-LJ) having an average molecular weight of <667 Da [38], low molecular weight fucoidan (LMWF) having main MW of 760 Da [39], and LMF having an average MW of 800 Da [40]. Finally, low molecular weight fucoidan extract (LMF) having a main MW of <500 Da, prepared by digesting crude fucoidan extracted from mozuku (*Cladosiphon novae-caledoniae* Kylin) with an abalone glycosidase, which was used in this study [41]. These two size groups of fucoidan derivatives have been examined for their health benefits as well as for their therapeutic effects and were shown to exhibit broad biological activities such as anti-tumor, antioxidant, anticoagulant, anti-inflammatory, and immune-modulatory effects in in vitro and in vivo studies [22,23,42,43,44]. With regards to its anti-tumor activity, fucoidan has been shown to exhibit suppressive effects in lung, breast, liver, colon, prostate, and bladder cancer cells [22,45]. In addition, our previous in vitro studies revealed that LMF (MW < 500 Da) can enhance the anticancer activity of chemotherapeutic agents (such as cisplatin, tamoxifen, and paclitaxel) [46] and it also demonstrated beneficial immunomodulatory effects in a clinical trial [23]. It has been shown that the low molecular weight fucoidan with a MW of 7.6 kDa exhibited higher intestinal absorptivity than the medium molecular weight fucoidan (MW 35 kDa) when assessed in the plasma and urine after oral administration in rats [36]. This result suggests that the lower molecular weight fucoidan derivatives are superior to the ones with higher MW in the intestinal absorptivity which is in line with our long-time notion of using LMF (mainly MW < 500 Da) for clinical application along with collecting supportive basic research data [23,46,47]. In order to further support these conclusions, the low molecular weight fucoidan derivatives mimicking the structural features of the genus *Cladosiphon* have been synthesized and tested for their anti-cancer activities. Results showed that one of the sulfated tetrafucoside synthetic derivatives could reduce MCF-7 and HeLa cell growth while showing no cytotoxic effect on normal WI-38 cells [48]. Moreover, it has been suggested that the most important factor that affects the biological activities of fucoidan is the branching degree of the fucoidan rather than its molecular weight, monosaccharides composition, and sulfate degrees [49]. Thus, it could be envisioned that an efficiently absorbed LMF (main MW < 500 Da) in the small intestine is perhaps also distributed throughout the intercellular environments, including tumor microenvironments, and such localized LMF may suppress cancer cell growth. These data together present the presently used LMF as an advantageous molecule over high molecular weight derivatives in further pursuing studies for basic research and clinical applications. 

The molecular mechanism of fucoidan action involves suppression of the growth factor-stimulated pathway involving Ras/Raf cascade. A large amount of data supports the fucoidan inhibition of phosphorylation of intermediary transducers such as ERK1/2, PI3K, Akt, mTOR, c-Jun, c-Fos, EGF receptor, and vascular endothelial growth factor (VEGF) [21,22,39]. 

Accumulated data suggest that Ras regulates several downstream pathways to upregulate PD-L1 expression, while fucoidan suppresses Ras-regulated downstream signal transducers that coincide with those of the PD-L1 expression pathways. However, none of the preceding studies, as well as the relevant computer literature searches, explored the possibility that fucoidan suppresses PD-L1 expression regulated by the Ras-activated pathways. Therefore, we focused on examining the regulatory effects of LMF for several growth factors, PD-L1/PD-L2 and RhoA/B. We then examined our hypothesis that LMF differentially regulates PD-L1 expression in cancer and normal cells.

## 2. Results

### 2.1. Suppression of Cancer Cell Growth by Low Molecular Weight Fucoidan Extract (LMF)

To confirm whether LMF specifically suppresses cancer cell growth, HT1080 fibrosarcoma cells and normal human fetal lung diploid fibroblast TIG-1 cells were treated with varying concentrations (0–500 µg/mL) of LMF for 48 h. WST-1 cell viability assay showed significant dose-dependent suppression of HT1080 cell growth by LMF (** *p* < 0.01, *** *p* < 0.001) (Figure 3A), while there was no statistically significant suppressive effect on TIG-1 cell growth up to 500 µg/mL (Figure 3B). Moreover, the response of HT1080 cells to concentrations lower than 100 µg/mL of LMF showed a similar dose-dependent suppressive effect with both WST-1 and Hoechst 33342 DNA staining assays (Figure 3C). In addition, Hoechst 33342 dye staining method was employed as this dye stains DNA of both live- and apoptotic-cells [50,51] resulting in the measurement of cumulative numbers of both cell types and allows the visualization of the cytocidal effect of LMF when compared with the WST-1 assay, which measures only live or metabolically active cell numbers [52,53]. The results showed a significant dose-dependent reduction in the cumulative cell numbers (Figure 3C). A comparison of IC_50_ of WST-1 as 40.1 µg/mL and IC_50_ of Hoechst staining as 73.7 µg/mL revealed that approximately 12% of HT1080 cells are apoptosis-induced cells by LMF. These results demonstrate that LMF at even lower concentrations suppresses cell growth dose-dependently, and approximately 12% of HT1080 cell apoptosis was observed to be induced by 36.6 µg/mL of LMF up to 100 µg/mL. 

### 2.2. Comparison of PD-L1 Expression Levels in Seven Cell Lines by Immuno-Fluorescent Staining

Prior to examining the effect of LMF on PD-L1 expression, we compared PD-L1 expression levels in seven cell lines, using A549 and PC-9 cells as controls for low and high PD-L1 expression, respectively [12,44,54]. As shown in Figure 4, HT1080 cells expressed 1.06-fold higher PD-L1 compared to PC-9 cells. TIG-1 cells expressed PD-L1 levels comparable to those of the control cells (PC-9). Other cell lines expressed PD-L1 levels similar to or higher than those of the A549 cells. Therefore, we selected the HT1080 cell line for further studies to assess the effects of LMF on PD-L1 expression.

### 2.3. LMF Differentially Regulates Programmed Cell Death-Ligand 1 (PD-L1) and PD-L2 mRNA Expression in HT1080 and TIG-1 Cells

We examined LMF regulation of PD-L1/L2 expression in HT1080 and TIG-1 cell lines. HT1080 cells treated with 1 and 10 µg/mL of LMF for 48 h showed a significant reduction in PD-L1 (all, ** *p* < 0.01) and *PD-L2* (1 µg/mL, * *p* < 0.05; 10 µg/mL, ** *p* < 0.01) mRNA levels (Figure 5A). TIG-1 cells treated with 10 and 100 µg/mL of LMF for 48 h showed significant increases in PD-L1 (10 µg/mL, *** *p* < 0.001; 100 µg/mL, ** *p* < 0.01) and PD-L2 (all: **** *p* < 0.0001) mRNA levels (Figure 5B).

### 2.4. Suppression of PD-L1 Protein Expression by LMF

To further test the effects of LMF on PD-L1 expression, we measured the PD-L1 protein levels using an anti-PD-L1 antibody to test whether *PD-L1* mRNA suppression reflects similar suppression in protein levels. Western blot analysis of LMF-treated and untreated HT1080 cells was performed as described in the Methods section, and this method has been used by us to see the effects of fucoidan on HT1080 cells as well as other cultured cells [47]. PD-L1 protein band intensities of LMF-treated cells were lower than that of the control cells (Figure 6A). Protein bands in Figure 6A were quantitated for statistical evaluations and results show a significant reduction in PD-L1 protein (*** *p* < 0.001, Figure 6B) in LMF-treated HT1080 cells. Therefore, the PD-L1 protein levels reduced proportionally with a reduction in mRNA and the results suggest that the upstream signal transducers are mostly affected for PD-L1 transcription. 

### 2.5. Suppression of Cell Surface PD-L1 Protein Expression by LMF

Western blot analysis showed reduced PD-L1 protein levels in the HT1080 cell protein lysates proportionally with a reduction in PD-L1 mRNA levels. However, this method detects total PD-L1 proteins including intracellular premature PD-L1. The mature form of PD-L1 protein is anchored to the cell membrane with two extracellular domains [6]. It is necessary to show the level of surface PD-L1 exquisitely, as it is performing the most important and integral role in the immune escape mechanism of cancer cells. For this, we assessed if the effects of LMF are due to changes in the cell surface PD-L1 protein levels using the anti-PD-L1 antibody and analyzed this using flow cytometry, according to the published method. This method has previously been used to measure the PD-L1 expressed on the surface of several cell lines such as A549, H1975, and others using PD-L1 antibody followed by the flow cytometry analysis using a FACScan instrument [12]. Fluorescence histograms showed a small shift towards the left in LMF-treated HT1080 cells (Figure 7A, red line) compared with untreated control cells (Figure 7A, blue line), suggesting reduced levels of membrane-bound PD-L1 protein. To confirm this effect, mean fluorescence intensity (MFI) per cell, derived from histogram data, is presented in Figure 7B. These results show significant reduction in cell surface PD-L1 protein in LMF-treated HT1080 cells compared with the control cells (** *p* < 0.01). Thus, the results show that the PD-L1 mRNA level corresponds with that of the total as well as cell surface PD-L1 protein levels, suggesting an effect on upstream signal transducers regulating PD-L1 transcription.

### 2.6. LMF Does not Interfere with the Binding of PD-1 with PD-L1

The binding of PD-1 and PD-L1 occurs between T-cells and APCs for a normal T-cell response [3]. PD-1 binding with the PD-L1 expressed on cancer cells is an important step to evade the host immune surveillance system. Considering this, if LMF binds to PD-1 or PD-L1, attenuation and eventual inhibition of the normal T-cell response will occur. Therefore, we examined whether LMF inhibits binding of PD-1 with PD-L1. For this, we used PD-1 neutralizing antibody as a positive control which, as expected, showed dose-dependent inhibition of PD-1 and PD-L1 binding (Figure 8A). When LMF replaced the positive control in the same experimental setup, it did not interfere with the PD-1 and PD-L1 binding (Figure 8B). Therefore, the results show that LMF does not interact with PD-1 or PD-L1 and suggest that its suppressive effect is due to regulation of the other cell surface proteins as well as intracellular transducers involved with expression of PD-L1 and other genes. 

### 2.7. LMF-Suppression of Epidermal Growth Factor Receptor (EGFR) and Vascular Endothelial Growth Factor (VEGF) Expression

Growth factors play a significant role in carcinogenesis and metastasis [2,55]. The effect of LMF on EGFR, VEGF, and cell surface-associated protein expression was tested in HT1080 cells treated with different concentrations of LMF (0, 10, 100 µg/mL) for 24 h and mRNA levels were measured. LMF suppressed both EGFR mRNA (10 µg/mL, * *p* < 0.05; 100 µg/mL, ** *p* < 0.01), and VEGF mRNA (10 µg/mL; ** *p* < 0.01, 100 µg/mL; * *p* < 0.05) in HT1080 cells (Figure 9).

### 2.8. Suppression of RhoA and Stimulation of RhoB Expression by LMF

The Ras superfamily contains Rho GTPases which are involved in many cellular processes that regulate various signal transduction cascades that affect gene expression, cell cycle progression, cell migration, and many other cellular events by cycling from GTP-bound active state to GDP-bound inactive state [56]. In addition to their fundamental functions, Rho GTPases such as RhoA and RhoB play a significant role in cancer progression and inflammation [57]. To assess the effect of LMF on RhoA and RhoB expression, HT1080 cells were treated with different concentrations of LMF (0, 10, 100 µg/mL) for 24 h and mRNA levels were measured. LMF suppression of RhoA gene transcription at 10 µg/mL (** *p* < 0.001) and 100 µg/mL (* *p* < 0.05) (Figure 10A) was significant. In contrast, fucoidan caused a significant (all, **** p* < 0.001) and dose-dependent (10 and 100 µg/mL) upregulation of RhoB gene transcription in HT1080 cells (Figure 10B). All the results obtained are summarized in Table 1.

## 3. Discussion

In the present study, we examined the effects of LMF on six gene products including PD-L1, PD-L2, EGFR, VEGF, RhoA, and RhoB, involved in immune evasion, angiogenesis, and malignant transformation, using the human fibrosarcoma HT1080 cell line. We reveal that LMF downregulated five gene products, while upregulating RhoB expression. 

One of the important determinants to improve the efficacy of cancer therapy is inhibiting cancer cell evasion from the host tumor immune surveillance system [8]. Various tumors express increased levels of PD-L1 to evade such host defense mechanisms and eventually facilitate tumor growth [2,4,8,9]. Moreover, the tumor growth in terms of Breslow tumor thickness in melanoma specimens correlated with the level of PD-L1 expression, and the survival rate of patients expressing a high-level of PD-L1 was statistically lower than that of low-expressing patients with stage II melanoma [5]. Therefore, we confirmed the effects of LMF on the downregulation of PD-L1 expression in cancer cells. We first examined PD-L1 mRNA levels in six cancer cell lines, HT1080, MCF-7, PC-9, NIH:OVCAR-3, PANC-1, and A549 cells, and in one normal cell line TIG-1. All cell lines express PD-L1 levels similar to or greater than that of the A549 cell line, a low-expressing control [12,58]. The PD-L1 mRNA levels expressed by TIG-1 cells were comparable to those of the high-expressing control cell line PC-9 [12,44,54]. This observation agrees with earlier reports on several human cells, including lung cells and various tissues, which express low- to high-levels of PD-L1 [3,59,60]. These data strongly suggest the importance of selective inhibition of PD-L1 expressed on cancer cells but not on normal cells.

Among the cell lines tested, the HT1080 cell line expressed the highest levels of PD-L1 mRNA and was a suitable model to test the suppressive effects of LMF with emphasis on PD-L1 expression at the mRNA and protein levels. Several inherent characteristics of the HT1080 cell line are known since its establishment in 1974 [61]. HT1080 cells contain both wild-type and mutant N-Ras alleles and are expressed at the protein level. The mutant N-Ras protein induces cell transformation in cells even though it is present in far less amounts compared to that of normal p21 N-Ras in parental HT1080 cells [62]. The mutant N-Ras protein maintains a chronically activated state due to constitutive GTP binding, thereby conferring the constitutive activation of Ras-dependent signaling pathways. Representative Ras-regulated pathways are the Raf, Rac1, RhoA, and PI3K signal cascades, which are known to regulate mitogenesis, motility and invasiveness, actin cytoskeletal architecture, and cell survival, respectively [19,63]. Among the intermediary transducers, activated Raf/MEK is necessary for the aggressive tumorigenic phenotype in HT1080 cells but not the PI3K/Akt pathway [18,19]. However, others reported that the activated mutant N-Ras protein constitutively stimulates the Raf/MEK and the PI3K/Akt pathways to induce PD-L1 gene expression in several cancer cell lines [6,17]. 

Although, these data suggest that PD-L1 expression could be driven by the Ras regulated pathways in HT1080 cells, direct data proving an association between the Ras regulated pathways and PD-L1 expression in HT1080 cells are not available to date. In the present study, we show significant suppression of mRNA, protein and surface protein levels of PD-L1 in HT1080 cells by LMF. These results show suppression of activated mutant N-Ras regulated pathways for PD-L1 expression and existing data show many transducers in the Ras regulated pathways being suppressed by fucoidan [20,21,22,39]. LMF down-regulation of PD-L1 expression via Ras pathways in HT1080 cells could be explained by the possible regulatory pathways proposed for PD-L1 gene expression in cancer cells. According to this model, growth factor (GF)-stimulated pathways such as Ras/Raf and Ras/PI3K drive PD-L1 expression [6]. Thus, these pathways for PD-L1 expression coincide mostly with the Ras-regulated pathways proposed in HT1080 cells, as above. Although, the mutant N-Ras protein is constitutively active in regulating the pathways involving Raf, PI3K, and Rac transducers, the normal N-Ras-regulated pathways activated by GFs should be evaluated separately because HT1080 cells express functional wild-type and mutant N-Ras proteins [62].

Normal N-Ras protein is activated by exogenous stimuli such as by the binding of EGF, VEGF, and platelet-derived growth factor (PDGF) with their receptors (EGFR, VEGFR, and PDGFR). These transmembrane receptors mediate GTP-binding with the normal N-Ras protein for activation, enabling regulation of downstream transducers [63]. Cellular activities including proliferation, growth, survival, angiogenesis, metastases, and motility are regulated by the activated EGFR, VEGFR, and PDGFR through pathways such as the mitogen-activated protein kinase (MAPK) pathway: Ras/Raf, the PI3K pathway: PI3K/Akt, and the JAK/STAT pathway: JAK/STAT [64]. Available data shows that EGF-stimulated EGFR upregulates PD-L1 expression through EGFR/PI3K/Akt, EGFR/Ras/Raf, and EGR/PLC-γ signaling pathways [2,12,65,66]. Moreover, tumor cells can evade the immune system or promote their own growth through increased EGFR expression and indoleamine 2,3-dioxygenase 1 or VEGF production [67]. Although ligand-bound-activated EGFR with increased protein expression was present in HT1080 cells [14,68], the supernatant collected from the HT1080 cell culture did not contain secreted EGF but contained moderate concentrations of TGF-α which correlated with pEGFR expression [14]. Thus, an autocrine loop of TGF-α activates wild-type EGFR as well as the downstream factors in HT1080 cells, as both EGF and TGF-α belong to eight members of EGF family ligands, and TGF-α is an EGFR ligand in cancer [69,70]. Similarly, it has been reported that anti-EGF receptor mAbs prevented EGF and TGF-α induced growth stimulation in many human cells [14,71]. Accumulated data suggest that inhibition of EGFR or VEGF could inhibit HT1080 cell growth. In this study, we report significant suppression by LMF of both EGFR transcription and cell growth, suggesting a mechanism to suppress PD-L1 expression through regulation of EGFR expression. A plausible explanation for EGFR suppression by LMF would be that fucoidan binds with EGF causing prevention of EGF-induced phosphorylation of EGFR [72,73]. In support of this, earlier reports showed that fucoidan from *Laminaria guryanovae* exerts a potent inhibitory effect on EGF-induced phosphorylation of EGFR in mouse epidermal JB6 Cl41 cells [72] and fucoidan KW derived from brown algae *Kjellmaniella crassifolia* caused significant reduction in the levels of phosphorylated EGFR and PKCα proteins in A549 cells [43]. Others have shown that fucoidan suppresses growth factor-stimulated pathways involving the Ras/Raf pathway and inhibits the phosphorylation of intermediary transducers such as ERK1/2, PI3K, Akt, mTOR, c-Jun, c-Fos, EGFR, and VEGF [21,22,39]. While future studies are needed to confirm whether LMF binds with TGF-α in reducing EGFR phosphorylation, it is probable to suggest that reduction in the EGFR mRNA levels by LMF decreases the translated and phosphorylated EGFR levels. Contrary to others and ours, an existing report shows that EGFR could not be detected in HT1080 cells and suggested that HT1080 cell oncogenic signaling through Ras/PI3K/Akt is independent of EGFR [74]. In addition, fucoidan exerted a potent inhibitory effect on EGF-induced phosphorylation of EGFR and phosphorylation of ERK and JNK under the control of EGF. Furthermore, EGF-induced c-*fos* and c-*jun* transcriptional activities were inhibited by fucoidan leading to inhibition of AP-1 activity and cell transformation induced by EGF [72]. 

VEGF is present in a number of cancer cell lines including HT1080 cells and is suggested to play a crucial role in HT1080 cell metastasis based on results obtained using the anti-human VEGF antibody, which reduced spontaneous lung micrometastasis [75]. Moreover, inhibition of tumor-secreted VEGF limits primary tumor growth of sarcoma cell lines by inhibiting host angiogenesis [76]. An inhibitory mechanism of angiogenesis by low molecular weight fucoidan (LMWF), which is distinct from LMF, inhibits hypoxia-induced reactive oxygen species (ROS) formation, HIF-1α expression, VEGF secretion, and downstream VEGFR2/PI3K/Akt/mTOR/p70S6K1/4EBP-1 cascade in human bladder T24 cancer cells, ultimately suppressing HIF-1α/VEGF transcription and angiogenesis [39]. VEGF upregulates both PD-L1 mRNA and protein expression [2,77]. Thus, suppression of both VEGF-enhanced angiogenesis and PD-L1 expression contributes to inhibiting cancer cell evasion. Supporting evidence shows that fucoidan extract inhibits expression and secretion of VEGF which suppresses vascular tubule formation in HeLa cells and suppressing invasion of HT1080 cells [41]. In the present study, because LMF caused significant suppression of both VEGF mRNA and PD-L1 mRNA/protein levels in HT1080 cells, we can extrapolate that cancer cells become more susceptible to the host defense system. The rationale behind the suppression of the PD-L1 level by fucoidan comes from the fact that fucoidan directly binds with VEGF, thereby preventing VEGFR phosphorylation [70,78]. Other growth factors, PDGF and PDGFR, are constitutively expressed in HT1080 cells. PDGF-A is constitutively secreted into the medium, where it binds to and activates PDGFR and downstream PI3K and Akt pathways [19]. Although we did not test the suppressive effect of LMF on PDGF expression, there are studies showing that fucoidan suppresses PDGF expression. Oligo-fucoidan suppresses PDGF-induced cell proliferation and induces G_1_/G_0_ cell cycle arrest in airway smooth muscle cells [79]. Therefore, LMF is likely to aid in HT1080 cell growth inhibition by suppressing PDGF expression. As mentioned thus far, LMF suppresses VEGF and EGFR expression in HT1080 cells to suppress Ras-regulated pathways for PD-L1/PD-L2 expression. 

The Ras superfamily contains a subgroup of Rho GTPases composed of 20 kinds of intracellular signaling molecules including RhoA and RhoB proteins. Constitutively expressed RhoA functions in tumor development. Indeed, RhoA and its downstream pathway is constitutively active in HT1080 cells. The present study shows significant suppression of RhoA mRNA levels by LMF, suggesting that LMF downregulates Ras-controlled oncogenic activities via the RhoA pathway [19]. In contrast, the role of RhoB is not clearly known, it acts as either an oncogene under some conditions or a tumor suppressor gene under others [80,81]. RhoB expression is induced by EGF, PDGF, and many other agents, while several factors including N-Ras and EGFR suppress RhoB promoter activity in NIH3T3 cells and human cancer cell lines derived from lung, pancreatic, and cervical tumors [82]. Fucoidan suppresses PDGF (RhoB inducer) expression indicating downregulation of RhoB expression by fucoidan [79]. Conversely, LMF suppresses EGFR (RhoB suppressor), and fucoidan suppresses phosphorylation of various Ras-regulated transducers. Together these data suggest that LMF/fucoidan could act as an upregulator of RhoB because LMF significantly and dose-dependently upregulated RhoB mRNA levels, supporting the role of RhoB as a tumor suppressor. Thus, LMF performs a significant role in antitumor effect by upregulating RhoB expression in several cancer types [80,81]. Suppressive effects by fucoidan on Ras-regulated pathways are presented in Figure 11.

An important finding that LMF neither inhibited PD-L1 expression nor growth of the TIG-1 cells suggests selective activity towards cancer cells expressing PD-L1. Similar to our results with normal cells, another report shows a lack of effect of fucoidan on normal human mammary epithelial cell growth [83]. These results collectively suggest that LMF/fucoidan treatment against PD-L1 expression is effective because of its selective action on cancer cells. However, the anti-PD-L1 antibody treatment may not be efficacious because not all cancer patients express higher PD-L1 [67]. Similarly, the suppressive effect on PD-L1 expression conferred by LMF/fucoidan may not be efficacious for patients who are not expressing higher PD-L1 [67]. Thus, as both methods rely on the expressed PD-L1 or its expression in cancer cells, these methods have a common limitation in using PD-L1 as a treatment target. On the other hand, in addition to PD-L1 suppression by LMF, extensive data exists demonstrating that fucoidan exhibits various biological activities including the suppressive modulation of the intracellular transducers that regulate proliferation, growth, survival, angiogenesis, metastases, and motility in cancer cells [22,36,39,42,43,84]. Therefore, LMF/fucoidan exhibits anticancer effects due to the broad biological activities independent of its anti-PD-L1 effect.

## 4. Materials and Methods 

### 4.1. Cell Lines and Cell Culture

Seven established human cell lines and their culture conditions are shown in Table 2. 

### 4.2. Fucoidan

The abalone glycosidase-digested LMF, commercially available as “Power fucoidan,” was generously donated for this study by the Daiichi Sangyo Corporation (Osaka, Japan). LMF was prepared as previously described [2]. Briefly, high molecular weight fucoidan extract from seaweed of *Cladosiphon novae-caledoniae* Kylin was purified to 85% purity and digested with glycosidases to obtain LMF. LMF consists of a digested small molecular weight fraction (72%, MW < 500 Da) and non-digested fractions (less than 28%, peak MW: 800 kDa). LMF consisted mostly of fucose (73%), xylose (12%), and mannose (7%). The ratio of sulfation was 14.5%.

### 4.3. WST-1 Assay (Cell Viability Assay)

WST-1 assay is commonly used to measure the number of live cells after treating the cells with culture containing test agents [53]. This assay was used to measure the cytocidal effects of LMF on HT1080 cells. 100 µL each of HT1080 (2 × 10^4^ cells/mL) and TIG-1 (4 × 10^4^ cells/mL) cells were plated in 96-well microplates and cultured for 24 h at 37 °C. The WST-1 measurement was performed according to the standard protocol described by the manufacturer. Briefly, the cells were exposed to various concentrations of LMF for 48 h. Spent medium was removed and cells were washed once with PBS. To each well, 100 µL of culture medium containing 0.5 mM WST-1 (Dojindo) and 0.02 mM 1-methoxy phenazine methosulfate (Dojindo) was added and incubated for 1 h at 37 °C. Cell metabolic activity was measured at 450 nm using a microplate reader (Tecan, Männedorf, Switzerland). 

### 4.4. Hoechst Staining (Cellular DNA Staining for Cell Number Determination)

Hoechst 33342 dye stains both live and apoptotic cell DNA [50,51]. It is possible to estimate the cytocidal effect of LMF by analyzing WST-1 and Hoechst Staining assays. The cells in the same microplates used for WST-1 assay were fixed with 4% formaldehyde solution (Wako) for 30 min at 25 °C. Cells were washed once with PBS and stained with 100 µL PBS per well, containing 2 µg/mL of Hoechst 33342 (Dojindo), for 30 min in the dark at 25 °C, according to the instruction manual. Fluorescence intensity (excitation, 360 nm and emission, 465 nm) was measured by the multimode plate reader, infinite F200 PRO (Tecan).

### 4.5. Immuno-Fluorescence Staining

Previous studies have used the Immuno-fluorescence staining method to see the effect of fucoidan on T24 cells [39]. The same method was also used to detect PD-L1 in the cultured cells [58]. Therefore, we used this method to detect PD-L1 expressed in seven cell lines. HT1080 and A549 cells at 2 × 10^4^ cells/mL, MCF-7, PC-9, NIH:OVCAR-3, and PANC-1 cells at 3 × 10^4^ cells/mL, and TIG-1 cells at 4 × 10^4^ cells/mL were prepared, and 100 µL of each cell line was seeded per well in a 96-well black plate (Greiner Bio-One International GmbH, Kremsmünster, Oberösterreich, Austria) and cultured for 24 h. Cells were fixed with 4% paraformaldehyde (Wako) for 15 min. After washing with PBS, cells were treated with blocking solution containing 0.1% Tween-20 with 1% BSA in PBS for 1 h. After washing three times with PBS, cells were treated overnight at 4 °C with the anti-PD-L1 Rabbit mAb (E1L3N, Cell Signaling Technology Co., Danvers, MA, USA) diluted 800-fold with blocking solution. Rabbit mAb IgG Isotype Control (DA1E, Cell Signaling Technology) was diluted 800-fold with blocking solution and used as an isotype antibody control. After washing three times with PBS, cells were treated for 30 min with the secondary antibody, anti-Rabbit IgG (H+L), F(ab’)2 Fragment (Alexa Fluor^®^ 555 Conjugate) (Cell Signaling Technology), and diluted 1,000-fold with blocking solution. Cells were washed one time with PBS and stained with 2 µg/mL of Hoechst 33342 for 30 min. Fluorescence intensity of each cell line was measured using IN Cell Analyzer 1000 (GE Healthcare UK Ltd., Buckinghamshire, UK). 

### 4.6. Quantitative Reverse Transcription Polymerase Chain Reaction (qRT-PCR)

Polymerase chain reaction (PCR) has been used broadly, to detect a specific gene transcript [85], and to quantitate the target transcript level [86,87]. Quantitative detection and analyses of the expressed gene transcripts have been advanced greatly by incorporating automated instrumentations. Seven gene transcripts were examined by fully utilizing the commercially available instruments and reagents, as described below. Five milliliters of HT1080 cells were seeded at a density of 5 × 10^4^ cells/mL per 60 mm dish (FALCON, Corning Inc., Corning, NY, USA) and cultured for 24 h. Spent medium was replaced with the medium containing various LMF concentrations (0, 1, 10, or 100 µg/mL). After 48 h of treatment, total RNA was isolated using a High Pure RNA Isolation Kit (F. Hoffmann-La Roche Ltd., Basel, Switzerland) following the manufacturer’s protocol. cDNA was synthesized using total RNA and ReverTra Ace qPCR RT Kit Master Mix with gDNA Remover (TOYOBO Co., Ltd., Osaka, Japan) and was used for RT-PCR templates. For specific gene transcript amplification, cDNA template, primer pairs (Table 3), and THUNDERBIRDTMSYBR^®^ qPCR Mix (TOYOBO) mixture were placed in a 96-well PCR plate (NIPPON Genetics Co., Ltd., Tokyo, Japan) in triplicate. Primer sequences for amplifying the transcripts of seven genes (EGFR, VEGF, RhoA, RhoB, PD-L1, PD-L2, and GAPDH), shown in Table 1, were purchased from Sigma-Aldrich Co. (St. Louis, MO, USA). Quantitative reverse transcription polymerase chain reaction (qRT-PCR) for seven genes was carried out using Thermal Cycler DiceTM Real Time System TP800 (Takara Bio Inc., Shiga, Japan). The thermal program was as follows: denaturation at 95 °C for 30 s, 45 cycles of denaturation at 95 °C for 5  s, annealing at 60 °C for 10  s, and extension at 72 °C for 20 s. Fluorescence values for each gene were normalized using GAPDH as an internal standard and used for statistical evaluation. 

### 4.7. Flow Cytometry (FCM) Analysis

Five milliliters of 3.1 × 10^3^ cells/mL of HT1080 cells were plated on 60 mm dish and incubated for 24 h. The cells were then treated with 5 mL of medium with or without 10 µg/mL of LMF for 5 days. Cells were dislodged from the dish by treating with 0.05% Trypsin-0.02% EDTA solution (Thermo Fisher Scientific) for 3 min at 37 °C. Cells were fixed with 4% paraformaldehyde for 10 min at 37 °C, collected, and treated for 1 h with the drawback Rabbit mAb (E1L3N) or with the corresponding Rabbit mAb IgG isotype control (DA1E) diluted 800-fold with incubation buffer. After washing three times with PBS, cells were treated for 30 min with the secondary antibody, anti-Rabbit IgG (H+L), F(ab’)2 Fragment (Alexa Fluor^®^ 488 Conjugate) (Cell Signaling Technology) diluted 500-fold with incubation buffer. Fluorescence intensity of each cell line was measured using Flow Cytometer EPICS XL System II-JK (Beckman Coulter, Inc., Brea, CA, USA) which integrates flow cytometry technology, thus enabling the study of cellular populations [92]. Data analyses were carried out using Flow Cytometry (FCM) analysis software, FlowJo (Version 4.6.2) (Tree Star, Inc., Ashland, OR, USA) and the results expressed as histograms and MFI. 

### 4.8. Western Blot Analysis

In our previous study, the Western Blot method was used to analyze the expression of Bcl-2 family proteins [47]. We used this method to detect PD-L1 in HT1080 cell lysates treated with LMF. HT1080 cells were cultured for 5 days in the medium supplemented with or without 10 µg/mL of LMF. Cells were washed three times with cold PBS and lysed for 5 min with 400 µL of protein extraction buffer (M-PER Mammalian Protein Extraction Reagent (Thermo Fisher Scientific) supplemented with 1% of Halt Protease Inhibitor Single-Use Cocktail (Thermo Fisher Scientific) and 5 mM EDTA). Lysates were collected in 1.5 mL microtubes and cell lysis was further ensured by repeated sonication for four times at 5 s intervals using the Handy Sonic model UR-20P (TOMY SEIKO CO., LTD., Tokyo, Japan). Protein extracts were centrifuged at 14,000× *g* for 15 min at 4 °C. Supernatants were recovered and stored at −80 °C. Prior to use, the protein concentration was determined using the Bradford protein assay. Supernatants were diluted 50-, 100-, 200-, and 400-fold with PBS. A standard solution was prepared using bovine γ globulin (1 mg/mL) by serially diluting with PBS from 90 to 0 µg/mL with 10 µg/mL decrement. To determine the protein concentrations in the dilution series, 100 µL each of the protein extracts and standard solutions were mixed with 100 µL of 2.5-fold diluted Bio-Rad protein assay dye reagent concentrate (Bio-Rad Laboratories, Inc., Hercules, CA, USA) in a 96-well microplate (Nunc, Thermo Fisher Scientific), and absorbance at 595 nm were measured using a microplate reader (TECAN). Protein concentrations in the treated HT1080 cell extracts were determined based on the standard curve derived from bovine γ globulin. Equal amounts of protein from each sample were separated by electrophoresis through SDS-PAGE and transferred to Amersham Hybond P polyvinylidene fluoride 0.45 membranes (GE Healthcare) using The Trans-Blot^®^ SD semi-dry transfer cell apparatus (Bio-Rad) following the manufacturer’s protocol. Membranes were blocked for 1 h at room temperature with gentle shaking in Tris-buffered saline containing 5% skim milk powder (Wako). Membrane was cut into two pieces so as to apply different antibodies for detecting closer molecular weight proteins of GAPDH (37 kD) and PD-L1 (40-50 kD) unambiguously. The membrane was incubated overnight at 4 °C with the primary antibody, anti-PD-L1 Rabbit mAb (E1L3N) (Cell Signaling Technology) or anti-GAPDH Rabbit mAb (D16H11) (Cell Signaling Technology), diluted 1000-fold or 4000-fold respectively, in Solution 1 of Can Get Signal^®^ Immunoreaction Enhancer Solution (Toyobo). The membranes were washed twice and incubated for 1 h at room temperature with the Anti-Rabbit IgG, horseradish peroxidase (HRP)-linked secondary antibody (Cell Signaling Technology), diluted 2000-fold with Solution 2 of Can Get Signal^®^ Immunoreaction Enhancer Solution. Chemiluminescence signals of protein bands obtained by ECL prime western blotting detection reagent (GE healthcare), as described by the supplier, were detected with LAS-1000 (FUJIFILM Corporation, Tokyo, Japan), which were quantified and numerated using image J (1.4.6r) software. 

### 4.9. Inhibitory Activity of LMF on PD-1:PD-L1 Binding 

Inhibitory activity of LMF on PD-1 and PD-L1 binding was examined using PD-1:PD-L1(Biotinylated) Inhibitor Screening Colorimetric Assay Kit (BPS Bioscience, San Diego, CA, USA) following the manufacturer’s protocol. Briefly, PD-1 protein was coated on a 96-well plate and added biotinylated PD-L1 to form complex of PD-1:PD-L1:biotin. The plate was treated with Streptavidin-HRP followed by the addition of a colorimetric HRP substrate to produce color. Thus, this set up will serve as a positive control. To this set up, the addition of PD-1 neutralizing antibody as a test inhibitor will reduce color production serving as a second positive control. When a prospective inhibitor such as LMF is added to this set up, color production is reduced only when LMF binds to PD-1. For ligand control, the well is not coated with PD-1 and for the blank control well, biotinylated PD-L1 is not included. After incubation, these set-ups produce appropriate colors, which can be measured at 450 nm using a microplate reader (TECAN).

### 4.10. Statistical Analysis

Each experiment was performed at least in triplicate and repeated three times. The results are presented as the mean ± standard deviation (SD) values. The difference between the two groups was analyzed using the two-tailed Student’s *t*-test, and differences among three or more groups were analyzed using one-way analysis of variance (ANOVA) with Tukey’s multiple comparisons. Statistical analysis was done using Mini StatMate (ATMS Co., Ltd., Tokyo, Japan). A value of *p* < 0.05 represents a significant difference (* *p* < 0.05; ** *p* < 0.01; *** *p* < 0.001).

## 5. Conclusions

Our findings revealed that LMF exerts its effect by specifically suppressing not only mRNA levels but also cytoplasmic- and surface-protein levels of PD-L1 in HT1080 fibrosarcoma cells. We also provide data for the downregulation of VEGF, EGFR, and RhoA mRNAs, and the upregulation of RhoB mRNA in HT1080 cells. Together, these data suggest that LMF could be used as a complementary agent in the treatment of various types of cancers that express high PD-L1 levels.

## Figures and Tables

**Figure 1 marinedrugs-17-00421-f001:**
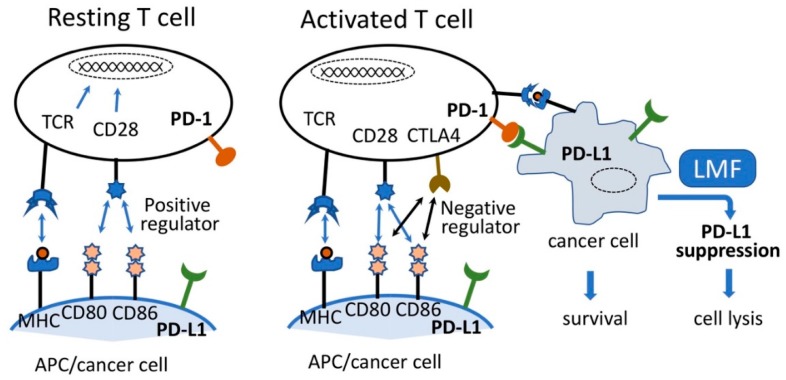
Simplified image of T-cell activation and cancer cell lysis. Resting T-cells are activated by the interaction of TCR:MHC and CD28:CD80 and CD86, leading to the expression of CTLA4. CTLA4 preferentially reacts with CD80 and CD86 causing activated T-cell lysis. Although imbalanced interaction of programmed cell death-1 (PD-1) on activated T-cell and programmed cell death-ligand 1 (PD-L1) on antigen-presenting cell (APC) causes T-cell lysis, such reaction between activated T-cells and cancer cells expressing PD-L1 will lead to the survival of cancer cells and facilitate cancer cell growth. Suppression of PD-L1 by low molecular weight fucoidan extract (LMF) leads to cancer cell lysis. Adapted from [7,13].

**Figure 2 marinedrugs-17-00421-f002:**
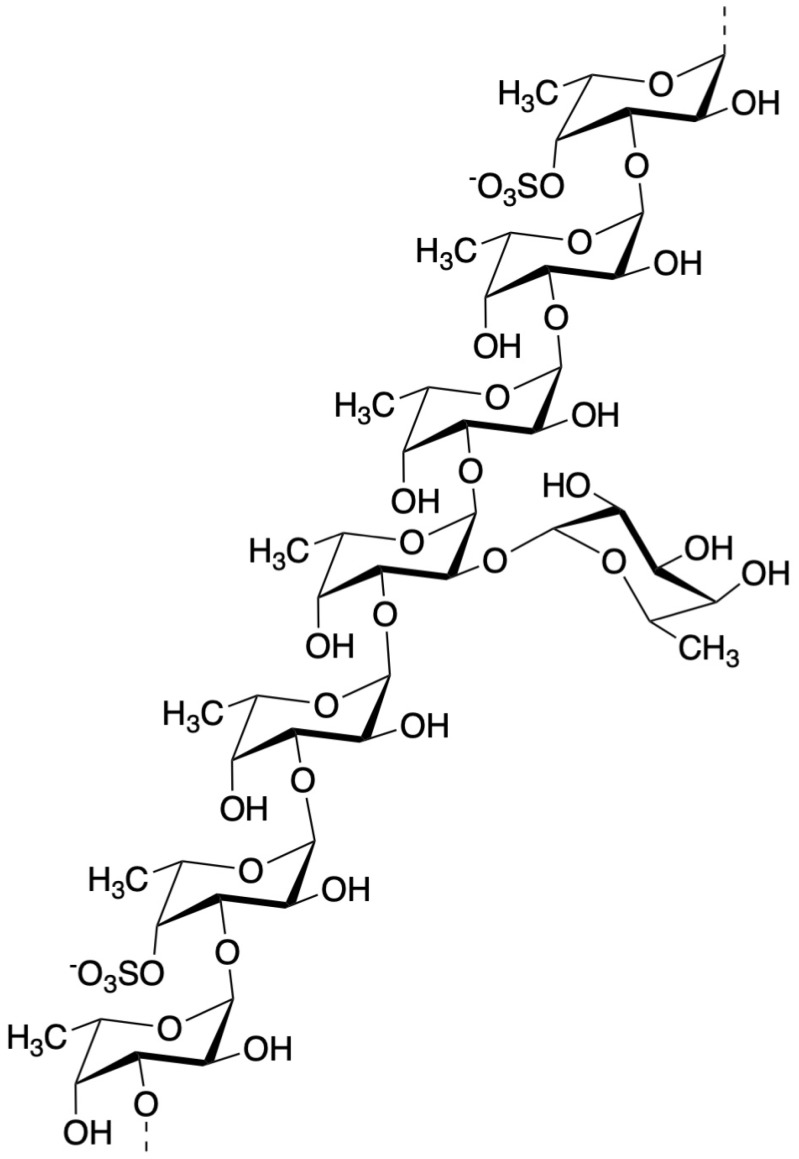
Structure of fucoidan from mozuku (*Cladosiphon okamuranus*) [27].

**Figure 3 marinedrugs-17-00421-f003:**
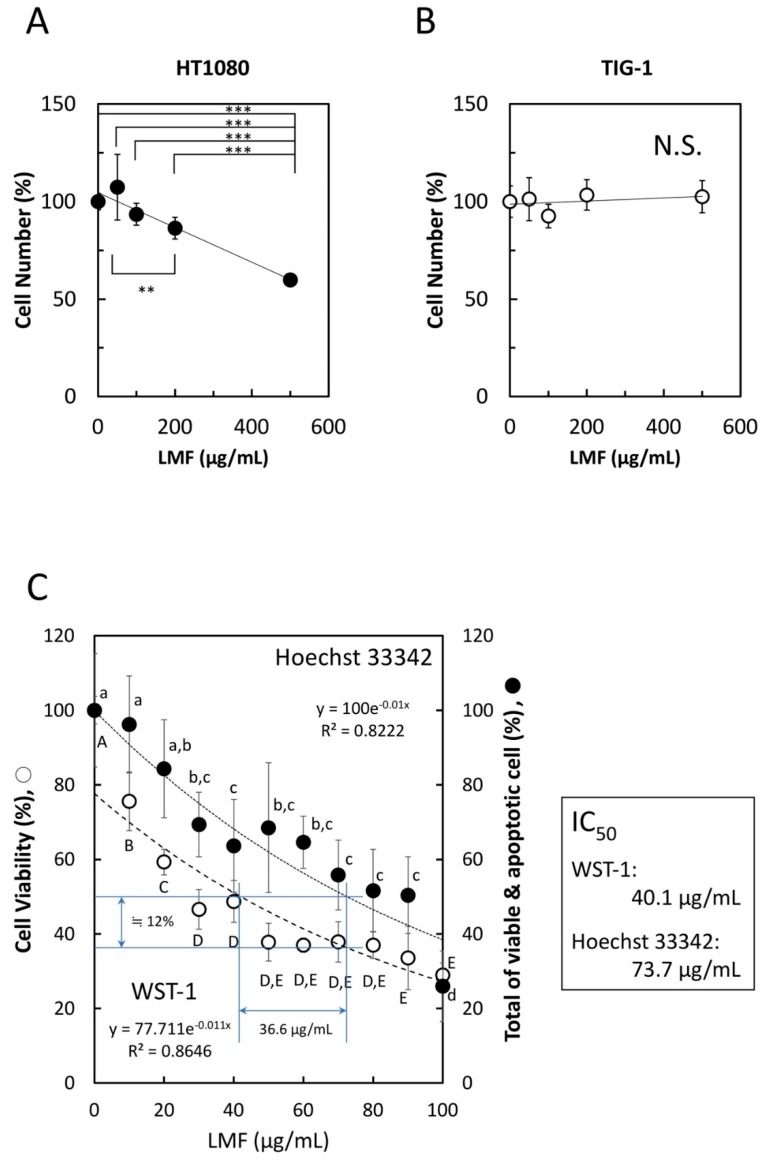
Effects of low molecular weight fucoidan extract (LMF) on cancer cell and normal cell growth. (**A**) HT1080 cells (1 × 10^4^ cells/mL) and (**B**) TIG-1 cells (3 × 10^4^ cells/mL), were plated (100 µL per well) in a 96-well microplate. After 24 h, varying concentrations (0, 50, 100, 200, 500 µg/mL) of LMF were added, the cells were incubated for another 48 h, and live cell numbers in each well were measured by the WST-1 assay. The measured values were used for statistical analyses by Student’s *t*-test. (**C**) Effect of lower LMF concentrations on HT1080 cell growth was examined using varying concentrations (0, 20, 40, 60, 80, 100 µg/mL) of LMF. After 48 h of treatment, cellular metabolic activity-based live cell numbers were measured by WST-1 (open circles, ○), and live cell numbers were measured by Hoechst 33342 DNA staining (closed circles, ●). The number of fucoidan untreated HT1080 cells was set to 100% to calculate the relative values of fucoidan-treated HT1080 cells. Statistical significance was determined by ANOVA with Tukey test. Different letters indicate statistically significant differences between each letter (*p* < 0.05). Half-maximal inhibitory concentration (IC_50_) values were calculated using approximation formulae.

**Figure 4 marinedrugs-17-00421-f004:**
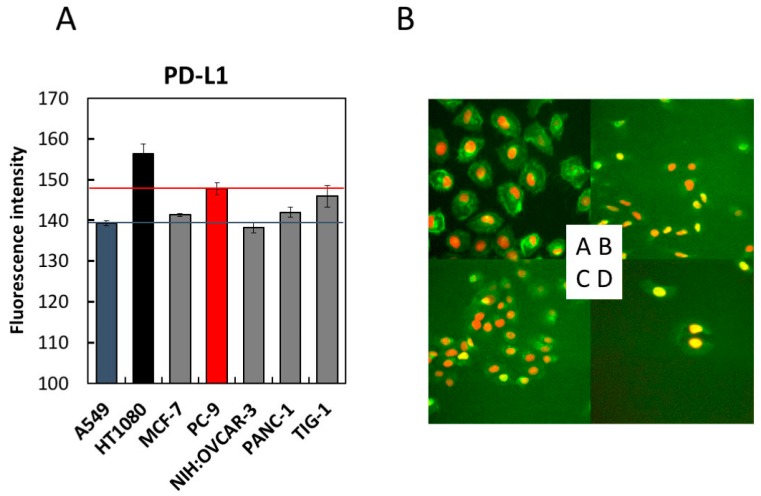
Comparison of programmed cell death-ligand 1 (PD-L1) expression levels in seven cell lines by immunofluorescence staining. Seven cell lines with indicated cell densities were prepared (HT1080 and A549 cells at 2 × 10^4^ cells/mL, MCF-7, PC-9, NIH:OVCAR-3, and PANC-1 cells at 3 × 10^4^ cells/mL, and TIG-1 cells at 4 × 10^4^ cells/mL) and seeded at 100 µL per well of a 96-well black plate followed by 24 h culture. PD-L1 detection was carried out using the anti-PD-L1 antibody, as described in the Methods section. Cells were stained with Hoechst 33342 for 30 min and analyzed using (**A**) IN Cell Analyzer 1000. The fluorescence intensities of each cell line were measured and converted to a numerical form for graphical presentation. For each cell line, six viewing fields per well were analyzed. A549 and PC-9 cells were used as low- and high-level controls for PD-L1 expression, respectively. (**B**) Four representative images acquired in (**A**) were shown. A, HT1080; B, A549; C, PANC-1; D, NIH:OVCAR-3.

**Figure 5 marinedrugs-17-00421-f005:**
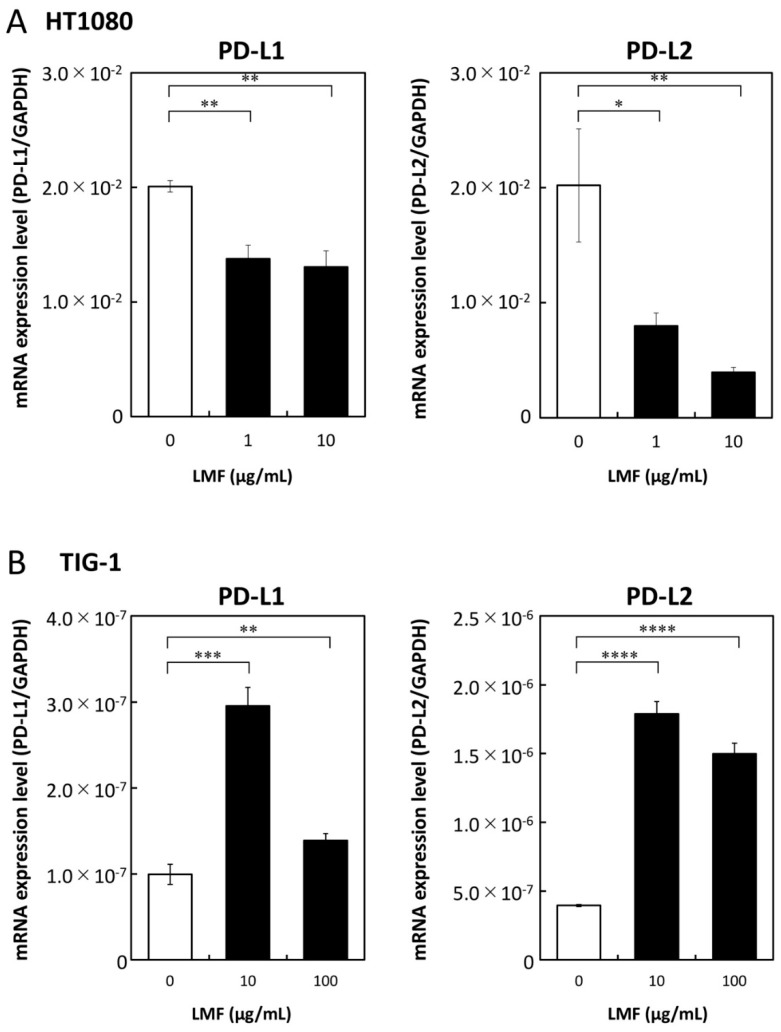
Low molecular weight fucoidan extract (LMF) differentially effects transcription of programmed cell death-ligand 1 (PD-L1) and PD-L2 mRNAs in cancer and normal cells. (**A**) HT1080 and (**B**) TIG-1 cells were treated with varying concentrations (0, 1, 10 µg/mL and 0, 10, 100 µg/mL) of LMF respectively, for 48 h. After treatment, RNAs were isolated and subjected to quantitative reverse transcription polymerase chain reaction (qRT-PCR) analyses. Values obtained from LMF-treated cells were compared with those of untreated cells using Student’s *t*-test (* *p* < 0.05; ** *p* < 0.01, *** *p* < 0.001; **** *p* < 0.0001, *n* = 3).

**Figure 6 marinedrugs-17-00421-f006:**
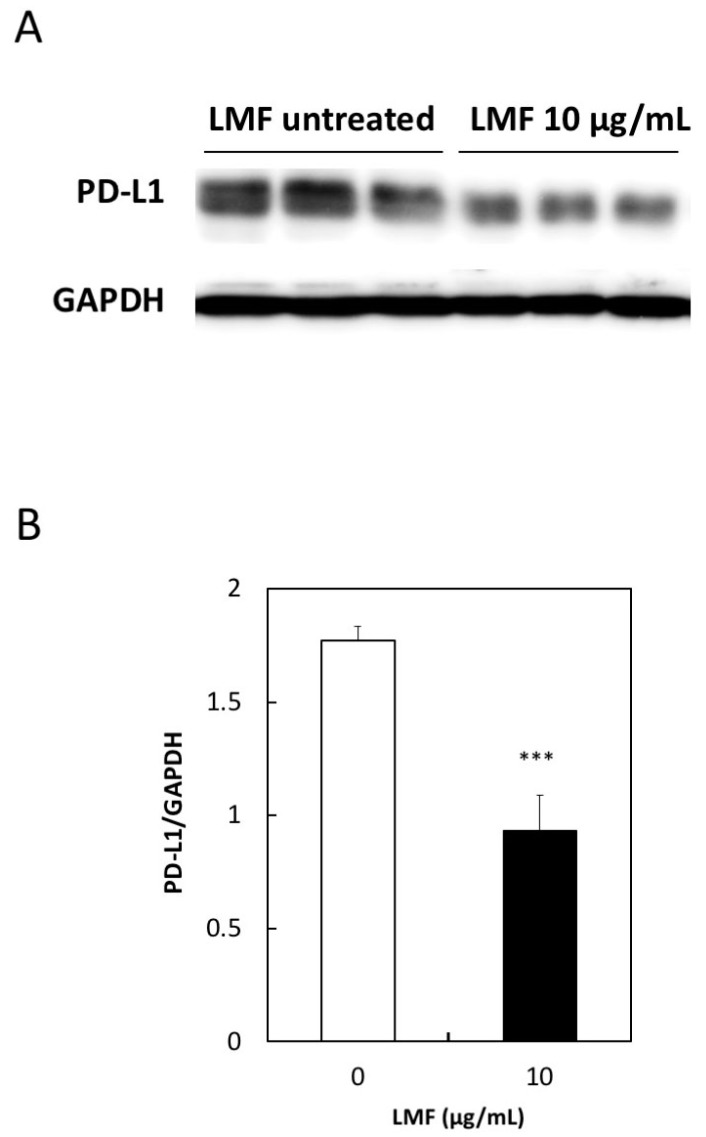
Low concentration of low molecular weight fucoidan extract (LMF) suppresses programmed cell death-ligand 1 (PD-L1) protein expression. (**A**) Western blot detection of PD-L1 protein in total protein lysates from HT1080 cells treated with 10 µg/mL of LMF for 5 days by replacing every 2 days with fresh media containing 10 µg/mL of LMF. Control untreated HT1080 cells treated similarly but without LMF. (**B**) Chemiluminescence protein bands from LMF-treated and untreated samples quantitated and bands from LMF-treated cells compared with that of untreated cells using Student’s *t*-test. (*** *p* < 0.001, *n* = 3).

**Figure 7 marinedrugs-17-00421-f007:**
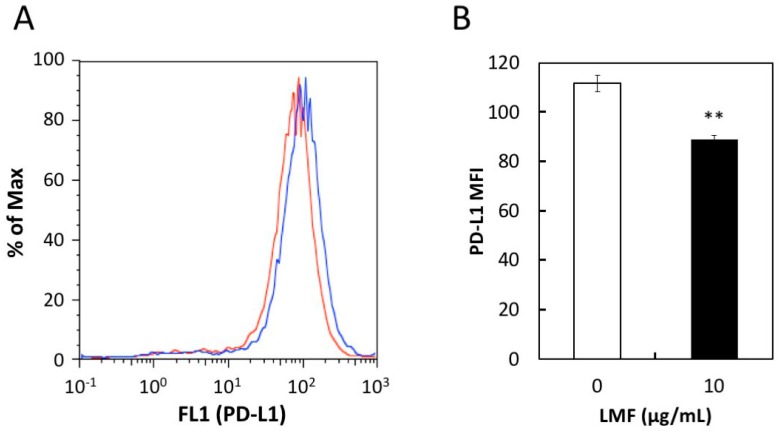
Effect of low amount low molecular weight fucoidan extract (LMF) treatment for 5 days on programmed cell death-ligand 1 (PD-L1) protein expression in HT1080 cells. HT1080 were treated with a low concentration of LMF (10 µg/mL) for 5 days and changes in PD-L1 protein levels were measured by flow cytometer. (**A**) PD-L1 protein expressed on the cell surface of HT1080 cells with (red line) and without (blue line) LMF treatment detected using anti-PD-L1 antibody and histograms were generated. (**B**) Mean fluorescence intensity (MFI) per cell was calculated using histogram data. Student’s *t*-test (** *p* < 0.01, *n* = 3) compares MFI of LMF-treated cells with that of untreated cells.

**Figure 8 marinedrugs-17-00421-f008:**
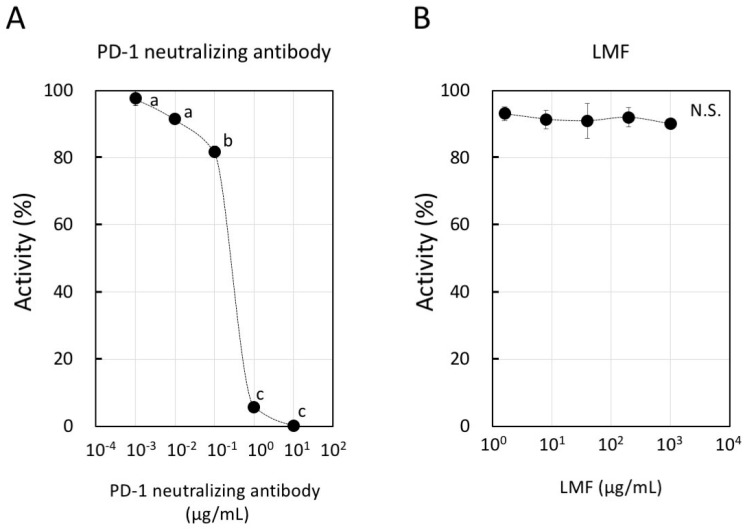
Inhibitory activity of programmed cell death-1 (PD-1): PD-L1 binding by low molecular weight fucoidan extract (LMF). (**A**) We used PD-1 neutralizing antibody as a positive control in dose-dependent inhibition of PD-1 and PD-L1 binding. Statistical significance was by ANOVA with Tukey test. The letters, a, b, and c indicate statistically significant differences between each letter (*p* < 0.05). (**B**) LMF was used to observe a dose-dependent inhibition of PD-1 and PD-L1 binding (N.S.: Not Significant).

**Figure 9 marinedrugs-17-00421-f009:**
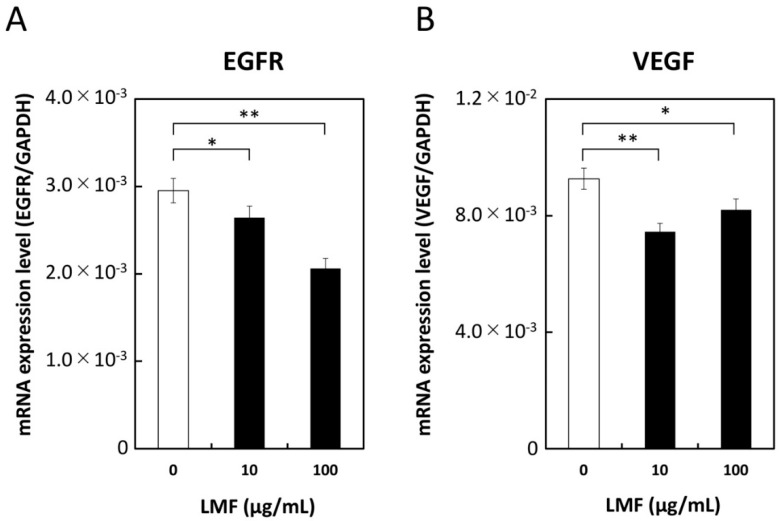
Suppression of epidermal growth factor receptor (EGFR) and vascular endothelial growth factor (VEGF) transcription by low molecular weight fucoidan extract (LMF). HT1080 cells were treated with 0, 10, and 100 µg/mL of LMF for 24 h. Total RNA was isolated and subjected to qRT-PCR analyses. LMF-treated cells were compared with untreated cells, and statistical evaluation was carried out using the Student’s *t*-test. (**A**) EGFR, (**B**) VEGF. (* *p* < 0.05, ** *p* < 0.01, *n* = 3).

**Figure 10 marinedrugs-17-00421-f010:**
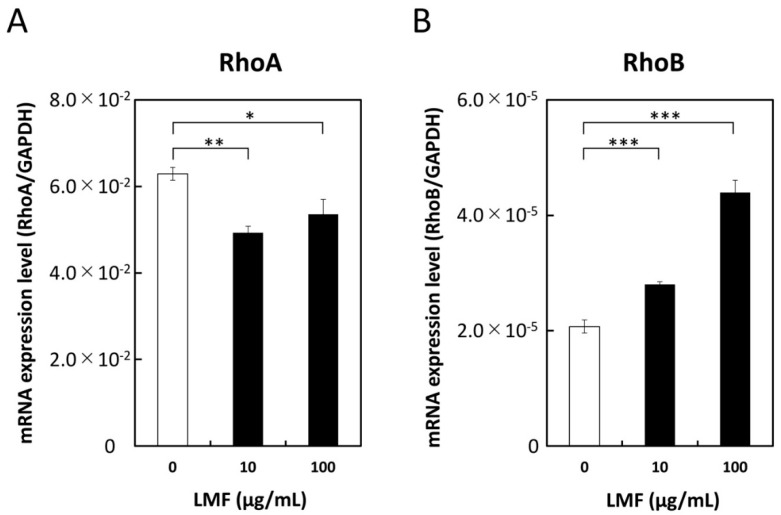
Low molecular weight fucoidan extract (LMF) suppresses RhoA and stimulates RhoB transcription. HT1080 cells were treated with 0, 10, and 100 µg/mL of LMF for 24 h. Total RNAs were isolated and subjected to qRT-PCR analyses. LMF-treated cells were compared with untreated cells and statistically evaluated using the Student’s *t*-test. (**A**) RhoA, (**B**) RhoB. (* *p* < 0.05; ** *p* < 0.01; *** *p* < 0.001, *n* = 3).

**Figure 11 marinedrugs-17-00421-f011:**
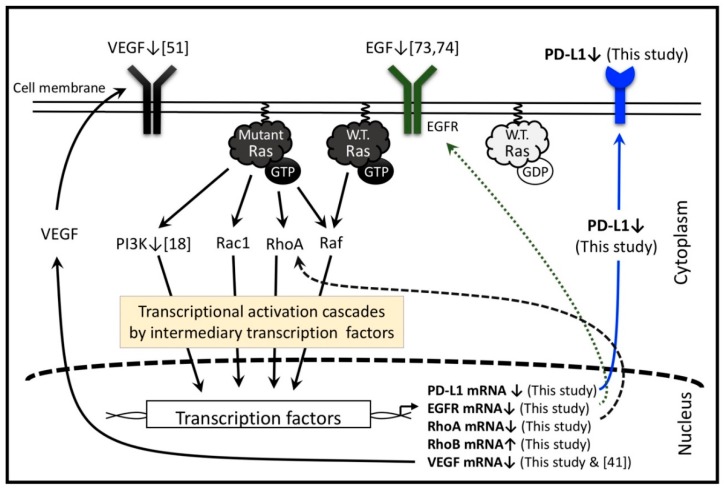
Low molecular weight fucoidan extract (LMF) and several fucoidan species regulate multiple transcription factors in the Ras-controlled pathways in various cells, including HT1080 cells. The numbers in the square bracket indicate the reference numbers, and downward and upward arrows indicate the factors up- and down-regulated by various fucoidan species reported in the references and LMF used in this study. Adapted from [2,6,19].

**Table 1 marinedrugs-17-00421-t001:** Summary of the results.

Cell Lines	Cell Growth Assay			PD-L1 Levels				mRNA			
				mRNA	Protein						
	Cell Count	WST-1	Hoechst 33342	+LMF *^1^	Total Surface			EGFR	VEGF	RhoA	RhoB
	+LMF				−LMF	+LMF	+LMF	+LMF			
HT1080	**↓**	**↓**	**↓**	**↓**	1.06	**↓**	**↓**	**↓**	**↓**	**↓**	**↑**
TIG-1	**→**	**–**	**–**	**↑**	0.99	**–**	**–**	**–**	**–**	**–**	**–**
A549	**–**	**–**	**–**	**–**	0.94	**–**	**–**	**–**	**–**	**–**	**–**
PC-9	**–**	**–**	**–**	**–**	1.00 *^2^	**–**	**–**	**–**	**–**	**–**	**–**
MCF-7	**–**	**–**	**–**	**–**	0.96	**–**	**–**	**–**	**–**	**–**	**–**
NIH:OVCAR-3	**–**	**–**	**–**	**–**	0.94	**–**	**–**	**–**	**–**	**–**	**–**
PANC-1	**–**		**–**	**–**	0.96	**–**	**–**	**–**	**–**	**–**	**–**
Figures	3A, B	3C	3C	5A, B	4A	6B	7B	9A	9B	10A	10B

* 1: PD-L2 mRNA was suppressed and increased by LMF in HT-1080 cells and TIG-1 cells, respectively. * 2: PC-9 cell mRNA level without LMF was set to 1.00 and the relative expression levels for other mRNAs were calculated. **–**: not tested.

**Table 2 marinedrugs-17-00421-t002:** Human cell lines and culture conditions.

Cell Lines	Registry No.	Types of Cells	Sources	Culture Conditions
HT1080	ATCC^®^ CCL-121™	Fibrosarcoma	ATCC^#1^	MEM supplemented with 10 mM HEPES, NEAA, 100,000 Units/L of Penicillin G Potassium, 0.1 g/L of Streptomycin sulfate and 10% fetal bovine serum (FBS) under a 5% CO_2_-humidified atmosphere at 37 °C.
A549	ATCC^®^ CCL-185™	Lung adenocarcinoma	ATCC^#1^	
MCF-7	ATCC^®^ HTB-22™	Breast adenocarcinoma	ATCC^#1^	
TIG-1	TKG0276	Normal fetal lung diploid fibroblast	IDAC^#2^	
PC-9	RCB4455	lung adenocarcinoma	RIKEN^#3^	DMEM supplemented with 10 mM HEPES, 100,000 Units/L of Penicillin G Potassium, 0.1 g/L of Streptomycin sulfate and 10% FBS under a 5% CO_2_-humidified atmosphere at 37 °C
NIH:OVCAR-3	RCB2135	Ovarian carcinoma	RIKEN^#3^	RPMI 1640 medium supplemented with 10 mM HEPES, 100,000 Units/L of Penicillin G Potassium, 0.1 g/L of Streptomycin sulfate and 10% FBS under a 5% CO_2_-humidified atmosphere at 37 °C.
PANC-1	RCB2095	Pancreatic carcinoma	RIKEN^#3^	

MEM (Eagle’s Minimum Essential Medium, Nissui Pharmaceutical Co., Ltd., Tokyo, Japan). HEPES (Dojindo Laboratories, Kumamoto, Japan), NEAA (Non-Essential Amino Acids, FUJIFILM Wako Pure Chemical Corporation, Osaka, Japan). Penicillin G Potassium (Meiji Seika Pharma Co., Ltd., Tokyo, Japan). Streptomycin sulfate (Meiji Seika Pharma). FBS: (fetal bovine serum, HyClone, Thermo Fisher Scientific Inc., Waltham, MA, USA). DMEM (Dulbecco’s Modified Eagle’s Medium, Nissui Pharmaceutical, Tokyo, Japan). RPMI (Nissui Pharmaceutical, Tokyo, Japan). ^#1^ ATCC: American Type Culture Collection, Manassas, VA, USA; ^#2^ IDAC: The Cell Resource Center for Biomedical Research, Institute of Development, Aging and Cancer, Tohoku University, Sendai, Japan; ^#3^ RIKEN: BioResource Research Center, Ibaraki, Japan.

**Table 3 marinedrugs-17-00421-t003:** Primer sequences used for qRT-PCR analyses.

Target Genes	F: Forward R: Reverse	Primer Sequences	References
*EGFR*	F R	5′-CGCAAAGTGTGTAACGGAATAG-3′ 5′-CCAGAGGAGGAGTATGTGTGAA-3′	[88]
*VEGF*	F R	5′-AAGGAGGAGGGCAGAATCAT-3′ 5′-ATCTGCATGGTGATGTTGGA-3′	[89]
*RhoA*	F R	5′-CCGGCGCGAAGAGGCTGGACT-3′ 5′-GCACATACACCTCTGGGAACT-3′	[90]
*RhoB*	F R	5′-GGTCCCCTGAGCATGCTTTCTGA-3′ 5′-GCCACACTCCCGCGCCAATCTC-3′	[90]
*PD-L1*	F R	5′-TATGGTGGTGCCGACTACAA-3′ 5′-TGCTTGTCCAGATGACTTCG-3′	[91]
*PD-L2*	F R	5′-TGACTTCAAATATGCCTTGTTAGTG-3′ 5′-GAAGAGTTCTTAGTGTGGTTATATG-3′	[91]
*GAPDH*	F R	5′-ATTGCCCTCAACGACCACTT-3′ 5′-AGGTCCACCACCCTGTTGCT-3′	[88]

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
