# Peer review of "Selective Suppression of Cell Growth and Programmed Cell Death-Ligand 1 Expression in HT1080 Fibrosarcoma Cells by Low Molecular Weight Fucoidan Extract"

_marinedrugs, 2019, doi:10.3390/md17070421_

Reviewer 1 Report

I feel the authors have addressed all my comments and requests and have introduced the necessary chnages and additions to the manuscript text.

Congratulations for the great work performed and for the acceptance of all suggestions as well as for the effort to clarify and correct where needed.

Author Response

Thank you for your comment and suggestion.

Reviewer 2 Report

 Manuscript ID: Kiichiro Teruya marinedrugs-548980

Title: Selective suppression of cell growth and programmed cell death-ligand 1 expression in HT1080 fibrosarcoma cells by low molecular weight fucoidan

Extract

Yes, it has been modified according to our suggestions.

I strongly recommend the paper for publication.

Kindest regards, Hesham

Author Response

Thank you for your comment and suggestion.

This manuscript is a resubmission of an earlier submission. The following is a list of the peer review reports and author responses from that submission.

Round  1

Reviewer 1 Report

This is a very well written paper with solid scientific support and relevant work for the natural products field in pharma (cancer) applications.

Introduction

The introduction is very much focused on the signaling pathways of cancer/tumour growth and very little on marine natural products/fucoidan, which is the goal of this journal.

I would advise, or would like to see, as a reader, some more background info on fucoidan and why it is relevant to use the LMF versus current format. (may take out some parts of the long discussion to reposition here).

Also, for the non tumour cell signalling experts it is quite difficult to follow some of the introduction parts without schemes/images.

Lines 54-64 - please rewrite this long sentence into smaller sentences and highlight the goal of this whole point. Alternatively build a figure to explain all this pathways and relations.

Results

This is a well rounded section. However, I would like to see some more original images of gels/cells and a lot more neghative controls on the pannels showed.

Figure 3 - mRNA supression - I would very much like to see the same pannel for the negative control GAPDH to see no suppression here.

Figure 4 - I would much like to see the whole gel to make sure that the control GAPDH and PD-L1 were run on the same gel under the same exact conditions... Just a cut of the bands is not sufficient to be evaluated.

Supression of cell surface PD-L1 protein expression:

Line 179 - this technique could have been further explained as to what is it really measuring and why was it chosen. These results are not clear to me neither is the selected method. There are other hystochemical/optical techniques that could have been used to compare the ammount of protein at the cell surface with and without LMF.

Figure 7 and 8 - again, I would have liked to see a negative control in these pannels, as this is relevant to show that this is a specific effect.

Discussion

Well discussed and related to current state of the art. Good and updated references.

A large portion of the discussion is theoretical and based on literature review, such as the potential in vivo outcome of LMF dosage in patients and its metabolic rate and path. I am not sure this is solid enough neither it is proprietary so I would afdvise to revise and reduce.

Reviewer 2 Report

Comments to Authors:

Title:

 The title is not good, so the authors would change title to more specific one.

Abstract:

Authors would mention the known active enzymes related to the work.

Authors would add the graphical abstract according to instruction of authors.

 Keywords:

Please do not use any abbreviation in the keywords list.

Introduction:

1.      The introduction of the manuscript is confusing and too long, please rephrase in a more concise form.

2.      Please mention the importance of marine natural product in drug therapy.

3.       “Fucoidan extract consists of poly saccharides but the authors describe it in the text as low molecular weight” please clarify and also give the structure.

4.      The authors could benefit from the following reference in the introduction:

M.A. Farag, M.I. Fekry, M.A. Al-Hammady, M.N. Khalil, H.R. El-Seedi, A. Meyer, A. Porzel, H. Westphal and L.A. Wessjohann (2017): Cytotoxic Effects of Sarcophyton sp. Soft Corals. Is There a Correlation to Their NMR Fingerprints? Marine Drugs 15, 211; doi:10.3390/md1507021.

Results and discussion:

1.      Please add and organize the results of the cell lines in the table.

2.      Authors would increase the references in this part.

3.      Authors would clarify why they selected the concentrations of 1 and 10 then 10 and 100 μg/mL of LMF.

4.      Cellular metabolic activity-based cell number measurement by WST-1 showed a half maximal inhibitory concentration (IC50) of 40.1 µg/mL, while live cell numbers measured by Hoechst staining showed an IC50 of 73.7 µg/mL. Authors would clarify this statement.

5.      “host tumor immune surveillance system” this sentence was repeated more than one time in this section.

6.      “However, LMF/fucoidan and the antibody treatment approaches toward the 413 PD-1/PD-L1 axis face a common drawback that not all cancer patients express higher PD-L1 [39].414 However, in this respect, extensive data exists, in addition to PD-L1 suppression….” Rewrite this part please.

Material and Method

  Cell lines and cell cultures could be organized in a table.

Experimental:

1.      Please add more references in this part.

2.      Authors would add the equation of the statistical analysis.

References:

Please check the style of references according to instruction of authors.

Figures:

Please improve the quality of figure 9.

Conclusion

  The authors would not repeat (and) in this sentence “mRNA, and cytoplasmic- and   surface-protein levels”.